

# Similarity-based metric analysis approach for predicting osteogenic differentiation correlation coefficients and discovering the novel osteogenic-related gene FOXA1 in BMSCs

Lingtong Sun[1,*], Juan Chen[1,*], Li Jun Li[1] and Lingdi Li[2]

[1] Hangzhou Xixi Hospital Affiliated to Zhejiang Chinese Medical University, Hangzhou, Zhejiang, China
[2] Department of Medical Oncology, Hangzhou Cancer Hospital, Hangzhou, Zhejiang, China
[*] These authors contributed equally to this work.

Corresponding authors
Li Jun Li, lilijun@zju.edu.cn
Lingdi Li, lilingdi2002413@126.com

## ABSTRACT

**Background**. As a powerful tool, bioinformatics analysis is playing an increasingly important role in many fields. Osteogenic differentiation is a complex biological process involving the fine regulation of numerous genes and signaling pathways.

**Method**. Osteogenic differentiation-related genes are collected from the online databases. Then, we proposed two indexes Jaccard similarity and Sorensen-Dice similarity to measure the topological relevance of genes in the human PPI network. Furthermore, we selected three pathways involving osteoblast-related transcription factors, osteoblast differentiation, and RUNX2 regulation of osteoblast differentiation for investigation. Subsequently, we performed functional a enrichment analysis of these top-ranked genes to check whether these candidate genes identified by similarity-based metrics are enriched in some specific biological functions and states. we performed a permutation test to investigate the similarity score with four well-known osteogenic differentiation-related pathways including hedgehog signaling pathway, BMP signaling, ERK pathway, and Wnt signaling pathway to check whether these osteogenic differentiation-related pathways can be regulated by FOXA1. Lentiviral transfection was used to knockdown and overexpress gene FOXA1 in human bone mesenchymal stem cells (hBMSCs). Alkaline phosphatase (ALP) staining and Alizarin Red staining (ARS) were employed to investigate osteogenic differentiation of hBMSCs.

**Result**. After data collection, human PPI network involving 19,344 genes is included in our analysis. After simplifying, we used Jaccard and Sorensen-Dice similarity to identify osteogenic differentiation-related genes and integrated into a final similarity matrix. Furthermore, we calculated the sum of similarity scores with these osteogenic differentiation-related genes for each gene and found 337 osteogenic differentiation-related genes are involved in our analysis. We selected three pathways involving osteoblast-related transcription factors, osteoblast differentiation, and RUNX2 regulation of osteoblast differentiation for investigation and performed functional enrichment analysis of these top-ranked 50 genes. The results collectively demonstrate that these candidate genes can indeed capture osteogenic differentiation-related features of hBSMCs. According to the novel analyzing method, we found that these four pathways have significantly higher similarity with FOXA1 than random noise. Moreover,
knockdown FOXA1 significantly increased the ALP activity and mineral deposits. Furthermore, overexpression of FOXA1 dramatically decreased the ALP activity and mineral deposits.

**Conclusion**. In summary, this study showed that FOXA1 is a novel significant osteogenic differentiation-related transcription factor. Moreover, our study has tightly integrated bioinformatics analysis with biological knowledge, and developed a novel method for analyzing the osteogenic differentiation regulatory network.

# INTRODUCTION

The widespread application of artificial intelligence is changing the paradigm of biomedical research. Different from traditional hypothesis-based research methods, artificial intelligence methods can automatically discover potential patterns and laws from data, providing new perspectives for revealing biological mechanisms (*Vora et al., 2023*). Bioinformatics analysis is a computer science method of learning and making predictions from data through algorithms and statistical models (*Saeys, Inza & Larrañaga, 2007*; *Xue et al., 2024*). The rapid development of bioinformatics analysis technology has brought new opportunities to biomedical research (*Sharma et al., 2021*; *Sharma et al., 2019*; *Holträter et al., 2020*). These methods, such as Similarity-based metric analysis, can extract valuable information from a large amount of complex biological data to provide support for disease diagnosis, prognosis prediction, treatment strategy optimization, *etc.* (*Tian et al., 2024*; *Jensen & Nielsen, 2024*). With the rapid accumulation of biological big data, how to use bioinformatics analysis to effectively analyze and utilize these data has become a key issue that urgently needs to be solved. This is of great significance for accelerating the progress of biomedical research and promoting the realization of precision medicine (*Halder et al., 2024*). Currently, there are some articles which combine bioinformatics analysis and experiments to study osteogenic differentiation (*Yu et al., 2024*; *Zhang et al., 2024*; *Feng et al., 2024*).

Osteogenic differentiation is an extremely complex biological process involving multiple cell types and regulatory mechanisms (*Valenti, Dalle Carbonare & Mottes, 2016*). This process begins with the differentiation of mesenchymal stem cells into osteoblasts, and goes through a series of finely regulated stages, ultimately forming functional bone tissue (*Infante & Rodríguez, 2018*). The regulatory mechanism of osteogenic differentiation involves the fine coordination of various cytokines, transcription factors, and signaling pathways (*Chan et al., 2021*). Among them, Wnt, BMP, Hedgehog and other signaling pathways play key roles in the proliferation, differentiation and maturation of bone cells (*Zhou et al., 2022*; *Hojo, Ohba & Chung, 2015*). In addition, transcription factors such as Runx2 and Osterix are major regulators of osteogenic differentiation (*Valenti, Dalle Carbonare & Mottes, 2016*; *Li et al., 2020*). Exploring key osteogenic differentiation factors

and understanding the molecular mechanisms of osteogenic differentiation not only help to deeply understand the biological process of bone development, but also provide an important theoretical basis for the treatment of various bone diseases such as osteoporosis and fractures (*Xu et al., 2023*).

FOXA1 is a member of the FOX family, commonly known as hepatocyte nuclear factor 3 $\alpha$. It regulates cell proliferation, development, differentiation, metabolism, aging, and other physiological functions (*Yang & Yu, 2015*; *Bernardo & Keri, 2012*). It can specifically bind to cis-acting elements in the promoter region of target genes, regulating downstream gene expression and exerting biological functions. However, its involvement in stem cell osteogenic differentiation is not well defined (*Sunkel et al., 2016*). Recent research has discovered that the FOX transcription factor family is linked to bone metabolism and a variety of bone illnesses, including osteoporosis, osteoarthritis, rheumatoid arthritis, intervertebral disc degeneration and bone malignancies (*Xu et al., 2021*; *Ye et al., 2018*).

In this study, we used bioinformatics analysis to develop a novel osteogenic differentiation regulatory network analysis model, aiming to automatically mine the core regulatory genes of this process from the database. In addition, we found the novel osteogenic-related gene FOXA1 in BMSCs.

## METHOD

### Data collection

The human PPI network including 19,344 genes is downloaded from the STRING database (*Szklarczyk et al., 2016*). It is then simplified by removing multiple edges and self-loops. Osteogenic differentiation-related genes are collected from online databases including GO, KEGG, BIOCARTA, PID, REACTOME, and WikiPathways. Among these databases, osteogenic differentiation-related gene sets or pathways are first retrieved using the keyword osteoblast. As a result, 10 osteogenic differentiation-related pathways including 337 unique genes are collected in our analysis. To determine the interaction between FOXA1 and osteogenic differentiation, four signaling pathways including hedgehog signaling pathway, BMP signaling pathway, ERK pathway, and Wnt signaling that have widely been studied to be associated with osteogenic differentiation are retrieved from The Molecular Signatures Databases (MSigDB) (*Subramanian et al., 2005*).

### Construction of similarity matrix between different genes

To calculate the similarity matrix between genes, we proposed two indexes Jaccard similarity and Sorensen-Dice similarity to measure the topological relevance of genes in the human PPI network. Given a graph $G(V, E)$, Jaccard similarity between any two vertexes can be defined as the following formula:

$$J(A, B) = \frac{|N(A) \cap N(B)|}{|N(A) \cup N(B)|} \tag{1}$$

where $N(A)$ is the neighbors of vertex A and $N(B)$ is the neighbors of vertex B. It calculates the proportion of co-neighbors among all neighbors of vertexes A and B in which a big similarity score indicates significant overlap of neighbors. In that way, the topological

similarity between any two genes can be caught and extracted by evaluating the overlap of co-neighbors.

Additionally, another similarity-based index Sorensen-Dice similarity is carried out to measure the topological similarity of genes which can be defined as:

$$SI(A,B) = \frac{2|N(A) \cap N(B)|}{|N(A)| + |N(B)|}.$$

(2)

Similar with Jaccard similarity, Sorensen-Dice similarity can utilize and calculate the topological similarity of genes by evaluating the topological overlap of neighbors. But it is normalized by dividing the total neighbors.

After evaluating the topological similarity of genes in the human PPI network based on Jaccard similarity and Sorensen-Dice similarity, these two indexes are then integrated them into a final similarity matrix which is defined as:

$$S(A,B) = \frac{J(A,B) + SI(A,B)}{2}.$$

(3)

It takes into consideration both Jaccard similarity and Sorensen-Dice similarity and calculates the average similarity which indicates the affinity of genes. This similarity matrix is then used for osteo-specific scoring in the following analysis. All of these analyses are conducted using the igraph R package.

## Functional enrichment analysis

To check whether these candidate genes identified by similarity index are enriched in some specific biological functions and states, we performed functional enrichment analysis of these genes in the GO and KEGG databases separately. We set 0.01 as the statistically significant threshold for BH-adjusted $p$-value. Pathways with BH-adjusted $p$-value less than 0.01 are considered to be statistically significant and presented by these candidate genes. GO enrichment analysis is performed in three different categories including biological process, cellular component, and molecular function. All of these analyses including functional enrichment analysis and visualization are implemented using clusterProfiler and enrichplot R packages (*Yu et al., 2012*).

## Permutation test

To check whether these osteogenic differentiation-related pathways can be regulated by FOXA1, we performed a permutation test to investigate the similarity score with four well-known osteogenic differentiation-related pathways including hedgehog signaling pathway, BMP signaling, ERK pathway, and Wnt signaling pathway. As a reference, we select 50 genes randomly from the whole gene list to calculate the average similarity score with FOXA1. This sampling process is repeated 10,000 times. Furthermore, the interactions of FOXA1 with these four pathways are also evaluated by averaging the similarity score with these genes involved in the osteogenic pathway. The similarity score between osteogenic differentiation-related pathways and those randomly selected gene sets are then compared to determine whether FOXA1 is significantly associated with these pathways which indicates a potential regulation mechanism during osteogenic differentiation.

## Cell culture and reagents

Cyagen Biosciences supplied hBMSCs (human bone mesenchymal stem cells, product number: HUXMA-01001, Cyagen Biosciences, Guangzhou, China), which may develop into osteoblasts, chondrocytes, and adipocytes under certain inductive circumstances. Adherent hBMSCs were cultured in culture flasks in a specific complete growth medium (HUXMA-90011; Cyagen Biosciences, Inc., Guangzhou, China) in a cell incubator at 37 °C with 5% $CO_2$ and passaged at around 80–90% confluence. Cells from passages two through six were utilized in future investigations.

## Lentiviral packaging and cell infection

Obio Technology (Shanghai, China) provided the hBMSCs with a lentiviral package that included lentiviral particles to overexpress (FOXA1 overexpress group, OE) and overexpress control particles (FOXA1 overexpress negative control group, OE-NC), knock down FOXA1 (FOXA1 knockdown group, KD), and knockdown control particles. When hBMSCs achieved 30–50% confluence, lentiviral particles containing 5 ug/ml polybrene were introduced to the growing medium per the manufacturer's instructions.

## ALP staining and quantitation

Cells grown in osteogenic induction media for 5 days in 24-well plates were washed three times with phosphate-buffered saline, then fixed with 4% paraformaldehyde (BOSTER, Wuhan, China) for 30 min at room temperature. The cells were stained with a BCIP/NBT ALP colour development kit (Beyotime). To evaluate ALP activity, an ALP activity assay (BOSTER, Wuhan, China) was used in accordance with product instructions.

## Alizarin Red S staining and quantitation

HBMSCs were cultivated in osteogenic induction media for 16 days after being passaged on 24-well plates. The cells were fixed in 4% paraformaldehyde (BOSTER, Wuhan, China) for 30 min at room temperature after being washed three times with PBS. Alizarin Red S solution (Cyagen Biosciences, Guangzhou, China)) was then added, and incubated at room temperature for 15 min. ARS stain was incubated with 10% cetylpyridinium chloride for 1 h at room temperature, then the solutions were collected, plated on a 96-well plate and measured at 560 nm with a microplate reader (ELX808; BioTek). The data were normalised to the control group.

## Statistical analysis

All statistical analyses were performed using GraphPad Prism v.7.0 (GraphPad Software, La Jolla, CA, USA). All experiments were done at least three times. The data is shown as the mean ± SD. When comparing two groups, statistical significance was established using a two-tailed Student's $t$-test, and when comparing more than two groups, one-way ANOVA followed by Tukey's post-hoc test was used. A $p$-value $<0.05$ was used to signify statistical significance.

## RESULT

### Similarity-based metric identifies osteogenic differentiation-related genes

To identify osteogenic differentiation-related genes, we performed similarity-based analysis to measure the topological relevance of genes involved in the human PPI network. It can be divided mainly into three steps: data collection, similarity measurement, and osteo-specific scoring. In the first step, human PPI network involving 19,344 genes is included in our analysis which then simplified by removing multiple edges and self-loops (Fig. 1B). Additionally, osteogenic differentiation-related genes are retrieved from published databases including GO, KEGG, BIOCARTA, PID, REACTOME, and WikiPathways (Fig. 1C). These genes are then used to extract osteogenesis-related features and grade genes into different levels. In the second step, we proposed two indexes Jaccard similarity and Sorensen-Dice similarity analysis to evaluate the topological similarity of genes in human PPI network (*Adamic & Adar, 2003*) (Fig. 1A). These two similarity indexes mainly concentrate on the overlap of neighbors of vertex A and B and measure the proportion of co-neighbors to evaluate the affinity of genes. After evaluating the similarity of genes using Jaccard and Sorensen-Dice similarity, they are then integrated into a final similarity matrix that presents the topological relevance of genes in human PPI network. In the third step, we mainly focus on these osteogenic differentiation-related genes and check whether some genes are significantly associated with these osteo-specific genes. We calculated the sum of similarity scores with these osteogenic differentiation-related genes for each gene which is then used to rank genes into different orders (Fig. 1D). This ranked gene list with osteo-specific score has potential in identification of osteogenic differentiation-related genes. We note that these genes with larger osteo-specific scores are more likely to participate in the regulation of osteogenic differentiation-related activity than other genes.

### Reliable of these candidate genes in regulating osteogenic differentiation of hBMSCs

A total of 337 osteogenic differentiation-related genes are involved in our analysis to grade genes into different levels. All of these genes are retrieved from osteogenic differentiation-related pathways. But whether these genes contribute equally to the osteo prioritization of genes is unclear. To identify these significant determinants of osteogenic differentiation-related genes, we selected three pathways involving osteoblast-related transcription factors, osteoblast differentiation, and RUNX2 regulation of osteoblast differentiation for investigation. We found that RUNX2, CTNNB1, JUN, and FOS are significant regulators of transcription factors (Fig. 2A). Analysis of the osteoblast differentiation pathway shows that BGLAP, SPP1, HOX, HGF, IGF1, BMP2, BMP4, SMAD3, and NOTCH1 are significant regulators (Fig. 2B). Analysis of RUNX2 regulation of osteoblast differentiation pathway shows that RUNX2, BGLAP, COL1A1, SRC, MAPK3, AR, MAPK1, and ABL1 are significant regulators (Fig. 2C). In summary, these analyses suggest that these genes are significant regulators of osteogenic differentiation and play significant roles in the prioritization and determination of osteogenic differentiation-related genes.

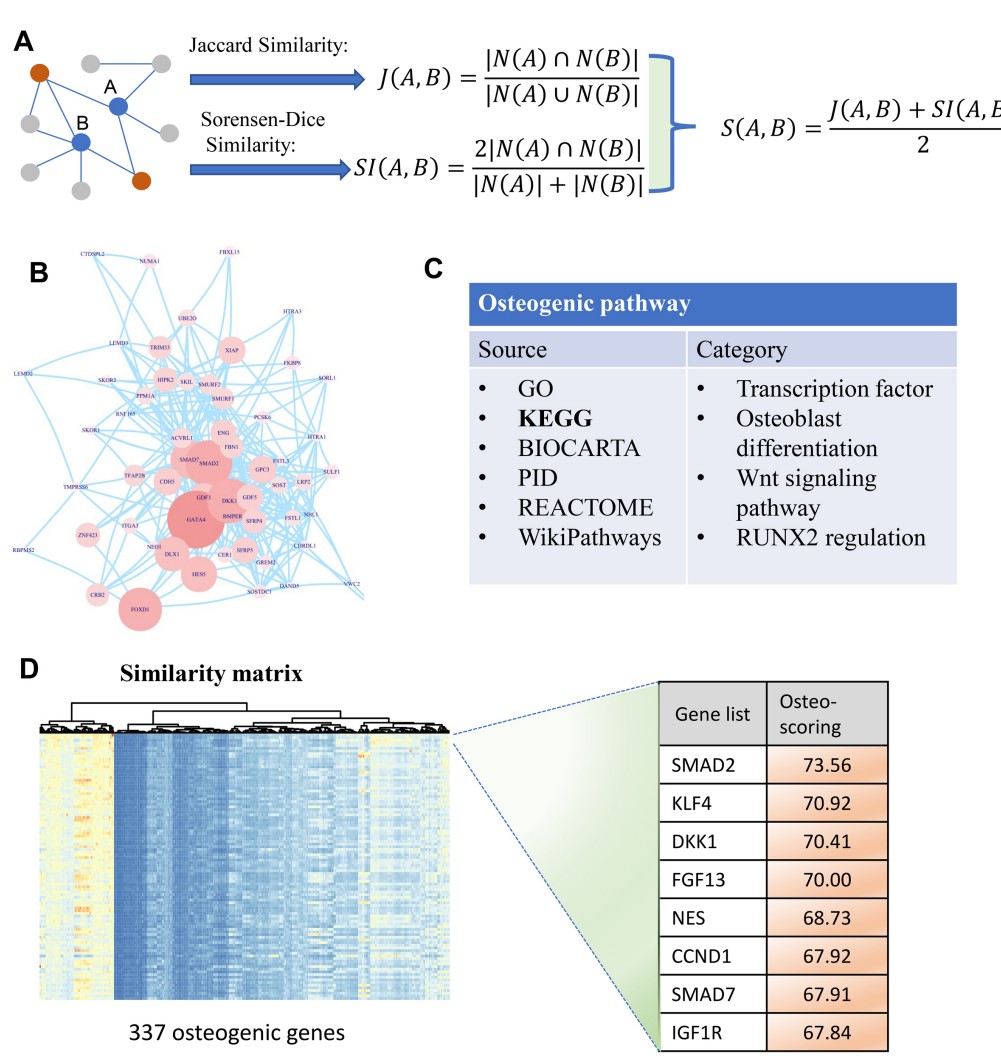

**Figure 1** **A diagram illustrating the identification of osteogenic differentiation-related genes.** (A) Two metrics Jaccard similarity and Sorensen-Dice similarity are carried out to measure the topological similarity of genes involved in a network. These two metrics are then integrated into a similarity matrix representing the affinity between genes. Red nodes are the co-neighbors of nodes A and B. (B) Human PPI network used to calculate similarity matrix between genes. (C) Collection of osteogenic differentiation-related pathways from different sources. (D) Similarity matrix with 337 osteogenic genes which are extracted from osteogenic differentiation-related pathways. This matrix is then used for osteo-specific scoring based on the similarity with osteogenic genes. The edges of B represent gene-gene associations.

To check whether these genes identified by similarity metrics can capture osteogenic features and are associated with osteogenic differentiation of hBMSCs, we investigated them in published literature. We found that almost all of these osteogenic differentiation-related genes lie in the upper gene list, suggesting that this ranked gene list can correctly prioritize and identify ossification-related genes. After removing these osteogenic differentiation-related genes, SMAD2, KLF4, DKK1, FGF13, NES, CCND1, SMAD7, IGF1R, TGFBR1, and KDR rank as the top 10 genes. Literature exploration shows that almost all of these

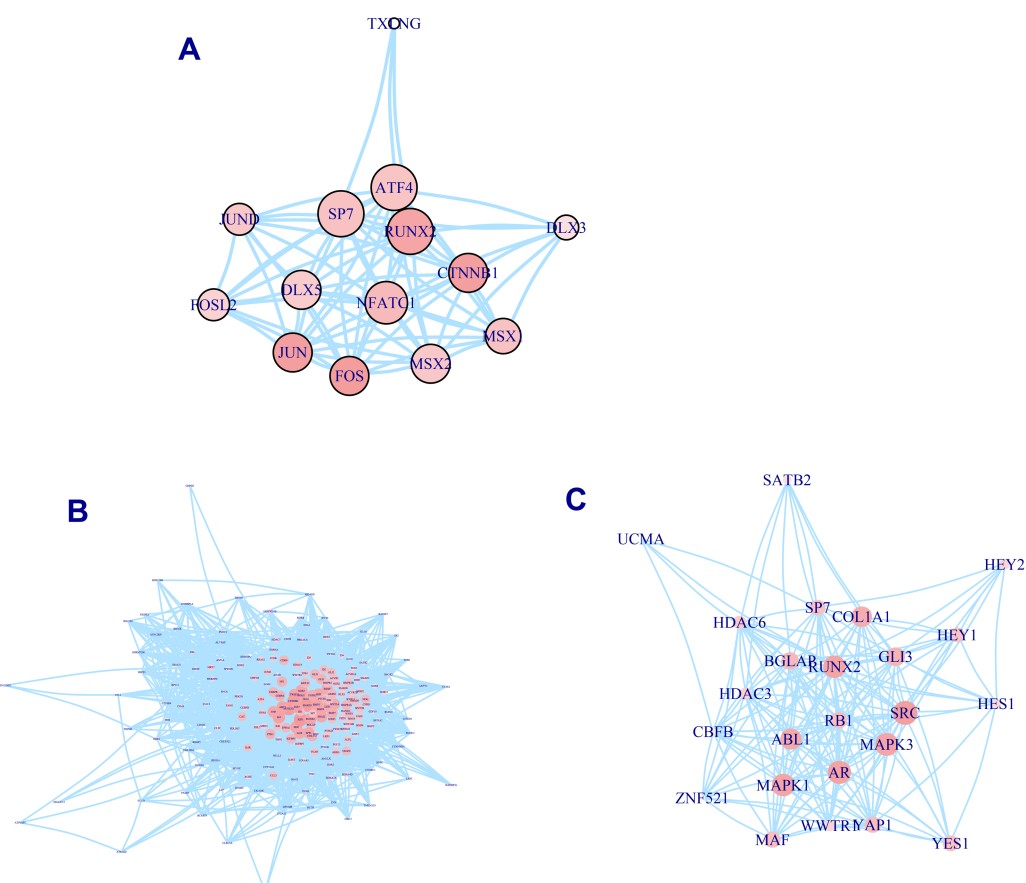

**Figure 2 PPI network of osteogenic differentiation-related gene set.** Transcription factor of osteoblast. (A) Osteoblast differentiation (B) and RUNX2 regulation of osteoblast differentiation (C) are presented in PPI network. Size of node is proportional with the similarity with top 100 candidate genes. Color of vertex is also positively correlated with the similarity of top 100 genes. The edges of A, B, C represent gene-gene associations.

genes have been reported to be associated with osteogenic differentiation. Activation of SMAD2 signaling pathway has been found to be associated with osteogenic differentiation (*Zheng et al., 2020*). KLF4, which encodes a protein that belongs to the Kruppel family of transcription factors, has been discovered to be a novel transcription factor of osteoblast differentiation (*Yu et al., 2021*). It has been found to regulate diverse cellular processes including cell proliferation, differentiation, and growth (*Ghaleb & Yang, 2017*). CCND1, which belongs to the highly conserved cyclin family, has been widely studied in various literature. Its downregulation has been observed to repress osteogenic differentiation and proliferation (*Wang & Cai, 2020*). Targeting Smad7 has been found to repress osteogenic differentiation of BMSCs (*Fang et al., 2019a*). The regulatory axis of GR/let-7f-5p/TGFBR1 has been discovered to be important for Dex-inhibited osteoblast differentiation (*Shen et al., 2019*). The IGF1R/PI3K/Akt signaling pathway has been shown to play a pivotal role in regulating osteogenesis (*Fang et al., 2019b*). In conclusion, these evidence suggests that

our similarity-based algorithm is reliable in the prioritization of osteogenic differentiation-related genes.

## Functional enrichment analysis of these candidate genes

To check whether these candidate genes identified by similarity-based metrics are enriched in some specific biological functions and states, we performed functional enrichment analysis of these top-ranked genes. The top 50 genes are selected for investigation of functional enrichment analysis. We performed functional enrichment analysis of these genes in GO and KEGG databases separately. GO enrichment analysis of these candidate genes shows that they are mainly enriched in osteogenic differentiation-related function including ossification and osteoblast differentiation (Fig. 3A). Additionally, other pathways such as mesenchyme development and mesenchymal cell differentiation are also presented by these candidate genes. Wnt signaling pathway, which has been widely studied and discovered to be associated with osteogenic differentiation, is observed to be enriched by these candidate genes. KEGG enrichment analysis of these genes shows that they are mainly enriched in cancer-related pathways including gastric cancer, hepatocellular carcinoma, breast cancer, and basal cell carcinoma (Fig. 3B). Apart from these pathways, the signaling pathway that regulates pluripotency of stem cells is also enriched in our analysis, suggesting significant value of these genes in regulating stemness. Other pathways such as hippo signaling pathway and wnt signaling pathway that have previously been reported to be associated osteogenic differentiation are also presented by these candidate genes. The top six enriched GO terms of these genes, which include cell fate commitment, epithelial to mesenchymal transition, mesenchymal cell differentiation, mesenchyme development, osteoblast differentiation, and regulation of animal organ morphogenesis, are presented in Fig. 3C. These results collectively demonstrate that these candidate genes can indeed capture osteogenic differentiation-related features of hBSMCs and confirm again that our similarity-based methodology is reliable in the identification of osteogenic genes.

## FOXA1 is a significant osteogenic differentiation-related transcription factor

Among these top 100 candidate genes, we found that some of them are transcription factors such as BMI1, FOXA1, SRF, MYC, FOXC2, CREBBP, and ZEB2. Literature mining of these transcription factors shows that most of these genes have been reported to be associated with osteogenic differentiation. FOXA1, which encodes a member of the forkhead class of DNA-binding proteins, has no literature reporting its interaction with osteogenic differentiation. Due to the significant rank of FOXA1 in the gene list, it is necessary and significant to investigate the potential value of FOXA1 in osteogenic differentiation.

To check whether FOXA1 can participate in the regulation of osteogenic differentiation-related activity, we selected four well-known pathways for exploration which include the hedgehog signaling pathway, BMP signaling pathway, ERK pathway, and Wnt signaling pathway that have widely been reported to be related to osteogenic differentiation. Permutation test is first performed to check whether FOXA1 is significantly associated with these four pathways. We found that these four pathways have significantly higher similarity
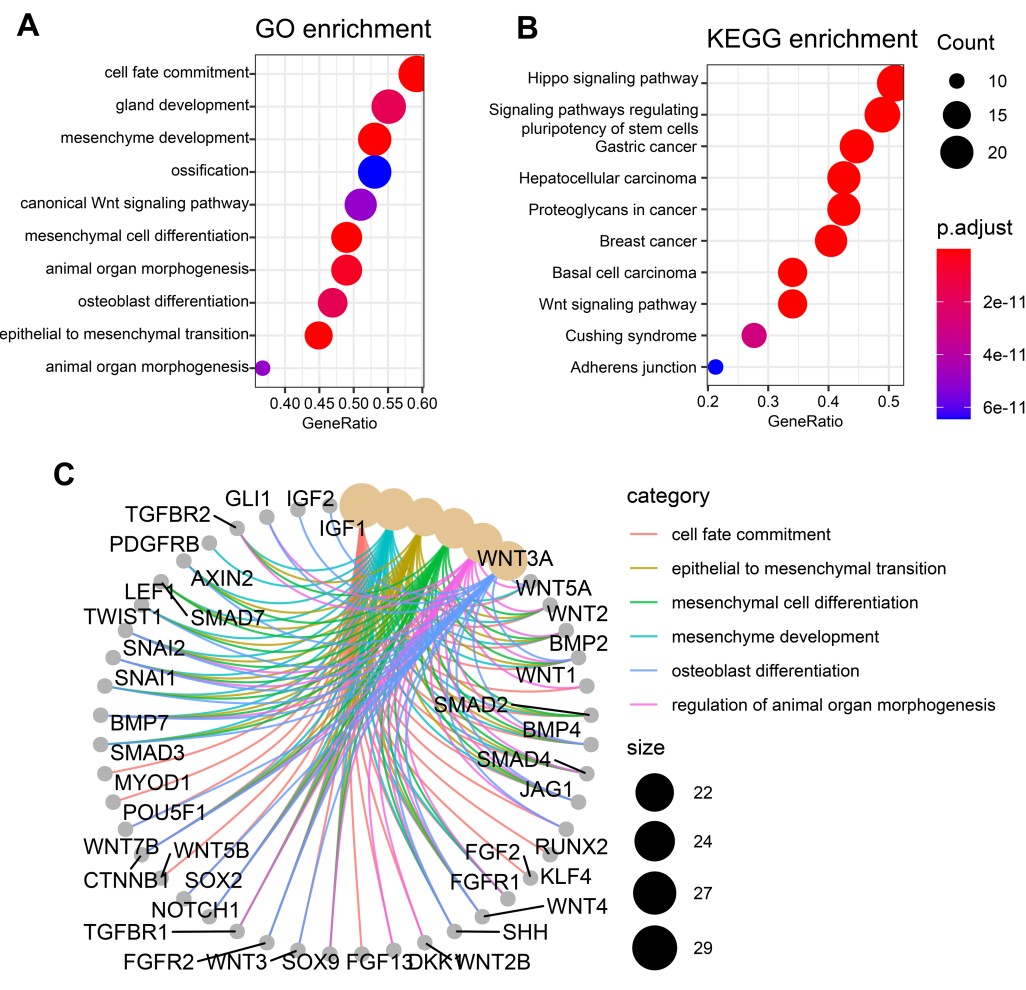

**Figure 3** **Functional enrichment analysis of top 50 candidate genes.** (A) GO enrichment analysis of these osteogenic differentiation-related candidate genes. (B) KEGG enrichment analysis of these candidate genes. (C) Top six enriched GO terms that are associated with these candidate genes. The edges of C represent gene-gene associations.

with FOXA1 than random noise (Fig. 4A), suggesting that FOXA1 is a significant regulator in mediating these osteogenic differentiation-related pathways.

We hypothesize that some genes involved in these pathways are more likely to be regulated by transcription factor FOXA1. In that way, FOXA1 can participate in the process of osteogenic differentiation-related activity by indirectly mediating these pathways. To identify target genes that are significantly regulated by transcription factor FOXA1, we analyzed the similarity score of these pathway genes with FOXA1. We found that GATA4, DKK1, SMAD2, and FOXD1 are significantly associated with FOXA1 in the BMP pathway with higher similarity scores than other genes (Fig. 4B), suggesting that these genes can be potentially regulated by FOXA1. Analysis of the Wnt signaling pathway shows that CCND2, DKK1, TCF7, HDAC2, NCOR2, and MYC are potential targets of FOXA1 (Fig. 4C). Analysis of the other two pathways hedgehog signaling pathway and erk pathway

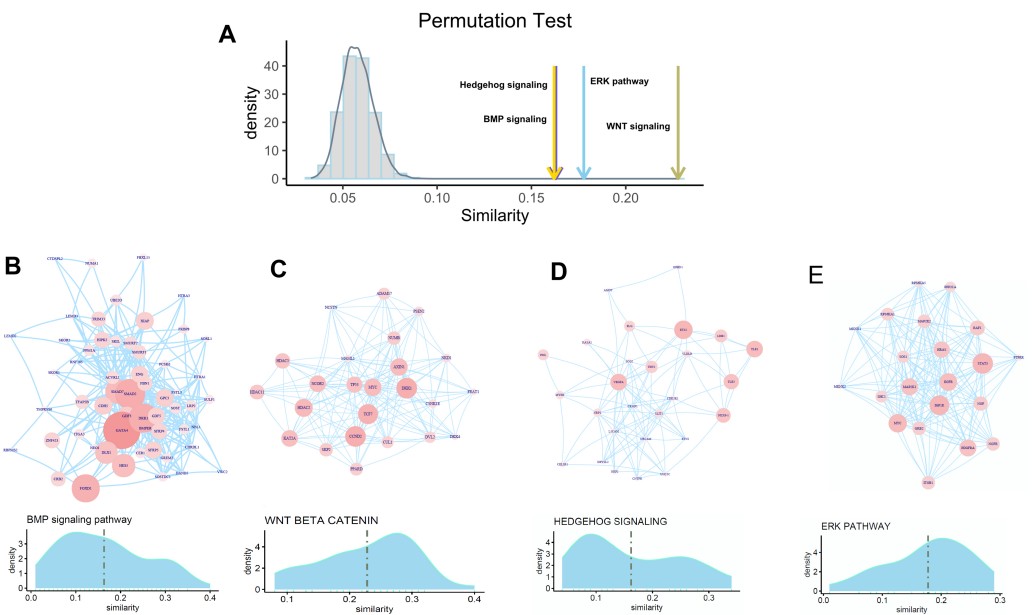

**Figure 4  Association of FOXA1 with osteogenic differentiation-related pathways.** (A) Similarity comparison of osteogenic differentiation-related pathways with randomly selected genes. Fifty genes are randomly selected from the total gene set which are repeated 10,000 times. Grey bar indicates the distribution of mean similarity of these 10,000 randomly selected gene sets. Similarity distribution and PPI network of four osteogenic differentiation-related gene sets including BMP signaling pathway, WNT beta catenin pathway, Hedgehog signaling pathway, and ERK pathway are presented in B, C, D, and E. The edges of B, C, D, E represent gene-gene associations.

shows that VEGFA, ETS2, TLE1, TLE3, and NKX6-1 are potential targets in the hedgehog pathway, and STAT3, EGFR, IGF1R, PDGFRA, and MAP2P1 are potential targets in the ERK pathway (Figs. 4D–4E). These analyses suggest that FOXA1 participates in the regulation of osteogenic differentiation by indirectly regulating these target genes of ossification-related pathways.

## FOXA1 knockdown enhanced alkaline phosphatase (ALP) activity and calcium deposit formation whereas FOXA1 overexpression decreased ALP activity and calcium deposit formation

ALP activity is a characteristic of early osteogenesis. On the 5 day of osteogenic differentiation, ALP activity was evaluated. ALP activity was significantly higher in the FOXA1-KD group compared to the KD-NC group ($p < 0.05$), and ALP staining results were similar (Figs. 5A and 5B). In comparison to the FOXA1 OE-NC group, the FOXA1 OE group had decreased ALP activity ($P < 0.05$) (Figs. 5A and 5B).

The calcium deposits were investigated using alizarin red staining (ARS). On day 16, the FOXA1 knockdown group had more calcium deposits than the KD-NC group, whereas the FOXA1 OE group had less calcium deposits than the OE-NC group. Quantification analysis yielded comparable results (Figs. 5A and 5B).

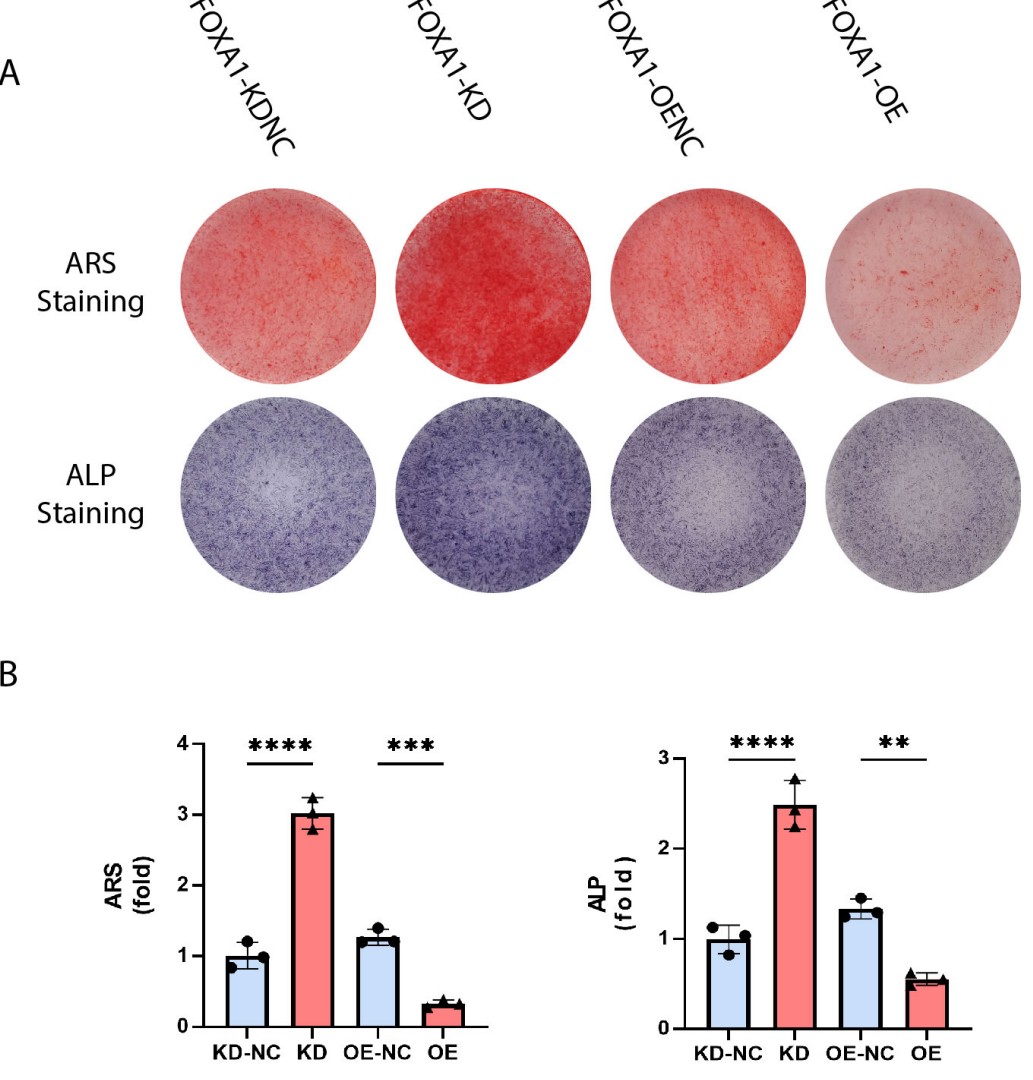

**Figure 5 Stain effects of FOXA1 knockdown and overexpress on osteogenic differentiation of hBM-SCs.** (A) Knockdown of FOXA1 significantly enhanced hBMSC ALP activity (after 7 days of osteogenesis) and calcium deposits. (after 14 days of osteogenesis) whereas FOXA1 overexpression decreased ALP activity and calcium deposit formation. (B) Quantitative detection of ALP and ARS. All data are means ± SDs ($n = 3$). * $p < 0.05$, ** $p < 0.01$, and *** $p < 0.001$ *versus* the control group.

## DISCUSSION

Currently, an increasing number of bioinformatics analysis research are focusing on the discovery of osteogenic differentiation-associated genes. However, practically all of them concentrate on the expression patterns of BMSCs and osteogenically generated samples (*Liu et al., 2021*; *Li et al., 2022*). Our study focuses on identifying osteogenic differentiation-related genes using similarity-based metric analysis and a human PPI network for gene classification by multiple online public database. This is different from the results of

the previous study (*Li et al., 2022*), in which identified shared osteogenic differentiation-related miRNAs and constructed an miRNA-transcription network between human BMSCs (hBMSCs) and human dental pulp stem cells (hDPSCs) on GEO database. Although the specific analytical methods and experimental conditions—including packages, database, analytical tools, experimental time, grouping and reagents, *etc.*—are different between our study and the previous study (*Li et al., 2022*), the transcription factor, FOXA1, was commonly identified as the crucial and novel osteogenic differentiation biomarker. The comparable results further demonstrate the reliability of our study, which indicate both studies provide credible and considerable findings about osteogenesis biomarkers. Therefore, it suggests these candidate biomarkers merit more study in subsequent studies.

The osteogenic differentiation of BMSCs is a highly regulated, multi-step, and complex physiological process that involves the post-transcriptional regulation of numerous miRNAs and transcription factors (TFs) (*Iaquinta et al., 2021*; *Gomathi et al., 2020*). Exploring the changes in genetic material during the osteogenic differentiation process of stem cells and selecting key genes for osteogenic differentiation as new gene targets are important scientific issues for accelerating bone repair and treating bone metabolic diseases (*Jiang et al., 2022*; *Ahmadi et al., 2022*; *Grottkau & Lin, 2013*). BMSCs are the source of osteogenic differentiation, possess self-renewal capabilities and the potential to differentiate into a variety of cell types, including osteoblasts,chondrocytes, and adipocytes (*Wang et al., 2016*; *Li et al., 2023*). As a key contributor to the bone formation, BMSCs are regulated by genetic factors (*Chan et al., 2021*; *Xu et al., 2023*; *Javed, Chen & Ghori, 2010*). Mesenchymal stem cells have been proved to be ideal seed cells for bone tissue engineering and are closely associated with bone defects fracture nonunion and osteoporosis (*Wang et al., 2016*; *Sun et al., 2022*; *Shang et al., 2021*; *Stamnitz & Klimczak, 2021*). Better understanding of the molecular mechanism in osteogenesis will enable researchers to design suitable targets for more effectively inducing bone tissue regeneration and treating related diseases (*Ansari, 2019*). Bioinformatics analysis enables us to explore the genetics alterations and identify novel biomarkers in mesenchymal stem cells osteogenesis.

In this study, we performed similarity-based analysis to measure the topological relevance of genes involved in the human PPI network, which divided mainly into three steps: data collection, similarity measurement, and osteo-specific scoring. These genes are found to be significantly associated with osteoblast-related pathways, suggesting a significant value of these genes in regulating osteogenic pathways. We found that these four pathways (hedgehog signaling, BMP signaling, ERK pathway, and Wnt signaling pathway) have significantly higher similarity with FOXA1 than random noise, suggesting that FOXA1 is a significant regulator in mediating these osteogenic differentiation-related pathways. Moreover, FOXA1 knockdown enhanced alkaline phosphatase (ALP) activity and calcium deposit formation whereas FOXA1 overexpression decreased ALP activity and calcium deposit formation (*Li et al., 2022*).

The human PPI network is included in our analysis. Jaccard similarity and Sorensen-Dice similarity are suitable for measuring the topological structure of genes in the network. But they are limited by the local features that only first neighbors are included in the similarity calculation without taking the second neighbors into consideration. The bioinformatics

analysis and vitro analyses show that FOXA1 is a newly discovered potential regulator in osteogenic differentiation of hBMSCs. However, new insights into the regulatory mechanisms involved is in need of further research. In addition, the effect of FOXA1 regulate BMSCs on osteoblast differentiation needs to be further investigated.

## CONCLUSIONS

Our research integrates bioinformatics analysis with biological knowledge to develop a unique approach utilizing the human PPI network and similarity-based metric analysis. This methodology investigates the regulatory network of osteogenic differentiation, providing a crucial theoretical foundation and potential targets for the treatment of various bone diseases. Furthermore, we identified FOXA1 as a novel and significant osteogenic differentiation-related transcription factor.

**List of abbreviations**

| | |
|---|---|
| **hBMSCs** | human bone mesenchymal stem cells |
| **ALP** | Alkaline phosphatase |
| **ARS** | Alizarin Red staining |

### Funding

Medical Health Science and Technology Project of Zhejiang Provincial Health Commission (Grant No. 2021KY939) funds and Doctoral Scientific Research Foundation of Xinxiang Medical University (505362) were received in support of this work. The authors declare no relevant financial activities outside the submitted work The funders had no role in study design, data collection and analysis, decision to publish, or preparation of the manuscript.

### Grant Disclosures

The following grant information was disclosed by the authors:
Medical Health Science and Technology Project of Zhejiang Provincial Health Commission: 2021KY939.
Doctoral Scientific Research Foundation of Xinxiang Medical University: 505362.

### Competing Interests

The authors declare there are no competing interests.

### Author Contributions

- Lingtong Sun conceived and designed the experiments, performed the experiments, analyzed the data, prepared figures and/or tables, and approved the final draft.
- Juan Chen conceived and designed the experiments, analyzed the data, authored or reviewed drafts of the article, and approved the final draft.
- Li Jun Li conceived and designed the experiments, performed the experiments, analyzed the data, prepared figures and/or tables, authored or reviewed drafts of the article, revision for important intellectual content, and approved the final draft.

- Lingdi Li conceived and designed the experiments, performed the experiments, analyzed the data, prepared figures and/or tables, authored or reviewed drafts of the article, revision for important intellectual content, and approved the final draft.

## Data Availability

The raw measurements are available in the Supplementary Files.

The osteogenic differentiation-related genes are available at the STRING database (https://string-db.org); GO (http://geneontology.org); KEGG (https://www.genome.jp/kegg/pathway.html); BIOCARTA (https://maayanlab.cloud/Harmonizome/dataset/Biocarta+Pathways); and PID (https://github.com/NCIP/pathway-interaction-database/tree/master/download, DOI: 10.1093/nar/gkn653);

REACTOME (https://reactome.org/download-data); WikiPathways (https://maayanlab.cloud/Harmonizome/resource/Wikipathways).

## Supplemental Information

Supplemental information for this article can be found online at http://dx.doi.org/10.7717/peerj.18068#supplemental-information.

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
