# Peer review of "Similarity-based metric analysis approach for predicting osteogenic differentiation correlation coefficients and discovering the novel osteogenic-related gene FOXA1 in BMSCs"

_PeerJ, doi:10.7717/peerj.18068_

## Round 0.1 · original submission · Major Revisions

Your manuscript has been reviewed by two experts in the field. As you can see from their comments, both raise many points that need to be addressed. In addition, Reviewer 2 feels that you should cite related papers more appropriately and clarify the novelty of your work among them. Please read their comments carefully and revise the manuscript accordingly.

Reviewer 1 ·

Basic reporting

please see no 4. (Additional Comments)

Experimental design

please see no. 4

Validity of the findings

please see no. 4

Additional comments

Title: Similarity-based metric machine learning approach for predicting osteogenic differentiation correlation coefficients and discovering the novel osteogenic-related gene FOXA1 in BMSCs
Authors: Sun et al.

1) Language and Typos
Please perform a thorough review to correct grammatical errors and improve readability. Specific issues include:
• Line 39: ‘Forthermore’ should be ‘Furthermore’
• Line 72: 'FOX' is used, while in Line 76, 'Fox' is used. Please ensure consistency.
• Line 208: ‘pathway)have’ should be ‘pathway) have’
• Line 209: ‘noise , suggesting’ should be ‘noise, suggesting’
• Line 210: ‘pathways.Moreover’ should be ‘pathways. Moreover’
• Line 279: ‘cells grown’ should be ‘Cells grown’

2) Abstract and Conclusion
The Abstract and Conclusion mention the use of ‘deep learning’ technology. However, the paper does not clearly describe the deep learning methods used or how they were applied. Please include a detailed discussion of the deep learning methods employed.

Additionally, while the Abstract refers to deep learning methods, no previous works on deep learning are cited. Consider citing recent works in this domain, such as DeepInsight (PMID: 3138803) and DeepFeature (PMID: 34368836).

The Conclusion is very abrupt and does not adequately summarize the key findings of the manuscript. Please provide a more comprehensive conclusion.

3) Methodology
Provide a rationale for the choice of similarity metrics used and discuss any limitations associated with these choices.

4) Cell Line Validation
Include more information on the validation of the hBMSC cell lines used in the study.

5) Permutation Tests
Provide more detailed information on the permutation tests conducted, including the significance levels used.

6) Figures
• Figure 4: Panels B, C, D, and E are difficult to read. Please improve their readability.
• From page 31 of the PDF onwards, the figures appear to be corrupted. Please ensure all figures are properly formatted and legible.

·

Basic reporting

This article uses clear and accurate English.
There are multiple terms of 'machine learning' and 'deep learning' in the introduction and discussion of this article. However, machine learning was not used in this analysis, so you must revise the terminology and content of discussion to 'bioinformatic analysis’.
If there are similar previous studies that have observed the new insights by applying bioinformatic analysis to osteogenic differentiation, you should be mentioned in the introduction.

Experimental design

The following aspects need to be revised and considered extensively.
1, The rationale for choosing Jaccard similarity and Dice similarity over other similarity metrics like cosine similarity is not sufficiently explained. The application of other similarity metrics, such as cosine similarity, is considered.

2, The text above the heatmap in Figure 1D should be removed.

3, The legends in Figure 1B do not explain the size and color intensity of the nodes. Additionally, the sentence "1934488 genes is included in our analysis" does not correspond to this figure.

4, The text on the nodes of the networks in Figure 2B, 4B, C, D, and E is illegible because it is too small and should be enlarged for readability. If the nodes in Figure 2B are too numerous to be displayed clearly, it would be beneficial to attach a supplementary table for reference.

5, The sentence described in lines 126-127 of the text "Literature investigation of other genes including SMAD7, IGF1R, and TGFBR1 all supports their interactions with osteogenic differentiation" is ambiguous and should be clarified to specifically describe how these genes are associated with osteogenic differentiation.

6, Figure 4A should statistically demonstrate the high similarity of the four osteogenic differentiation-related gene sets, which include the BMP signaling pathway, WNT/β-catenin pathway, Hedgehog signaling pathway, and ERK pathway. For example, after calculating the similarity of gene sets from other pathways and ranking them based on similarity, a Wilcoxon rank-sum test can be performed to statistically show that the similarity of the mentioned pathways ranks higher.

7, Figure 5 should quantify the expression level of FOXA1 knockdown under this experimental condition using qRT-PCR, and show in the text or included as supplementary data.

8, The cells stained with ARS in Figure 5A show some aggregation. You should seed the cells uniformly and quantify them to ensure accurate results.

9, The legends for all network diagrams lack descriptions of the edges.

Validity of the findings

The article(Li et al., Stem Cell research &Therapy(2022))demonstrated that FOXA1 silencing promotes the osteogenic differentiation of BMSCs via the ERK1/2 signalling pathway, and silencing FOXA1 in vivo effectively promotes bone healing, suggesting that FOXA1 may be a novel target for bone healing. You should discuss the differences between Figure 4A of the referenced article and Figure 5A of your study, as well as the similarities in the experimental design and results. Although the exact experimental conditions may differ, it is essential to cite the referenced paper and discuss the uniqueness of your study.

---

## Round 0.2 · Minor Revisions

The same two reviewers have reviewed your revised manuscript. As you can see both request additional minor revisions. Please read their comments carefully and revise the manuscript accordingly.

Reviewer 1 ·

Basic reporting

please check below

Experimental design

please check below

Validity of the findings

please check below

Additional comments

Thanks for revising the paper. I still have some comments which may be treated as minor comments.

1) The "Conclusion" Section (pg 17) is still very abrupt. It would be beneficial to provide more details about the work and the outcome achieved. Furthermore, what are the potential benefits. Currently, it is mentioned in the conclusion section that "built a unique approach to investigate the osteogenic differentiation regulatory network". However, it is not clearly articulated in which was the technique is unique. Perhaps some more description with appealing points in the conclusion section would help.

2) In the response to Methodology section the authors discussed the limitation in their response documents, however, it would be beneficial to include it in the manuscript as well.

·

Basic reporting

no comment

Experimental design

no comment

Validity of the findings

Reference 2 (McGovern et al., (2024)) is not an appropriate citation for bioinformatics analysis. It should be replaced by the review article (e.g., https://doi.org/10.1093/bioinformatics/btm344).

---

## Round 0.3 · accepted · Accept

I confirmed that you have appropriately revised the manuscript following the reviewers' comments